# Proteome Based Approach Defines Candidates for Designing a Multitope Vaccine against the Nipah Virus

**DOI:** 10.3390/ijms22179330

**Published:** 2021-08-28

**Authors:** Mohamed A. Soltan, Muhammad Alaa Eldeen, Nada Elbassiouny, Ibrahim Mohamed, Dalia A. El-damasy, Eman Fayad, Ola A. Abu Ali, Nermin Raafat, Refaat A. Eid, Ahmed A. Al-Karmalawy

**Affiliations:** 1Department of Microbiology and Immunology, Faculty of Pharmacy, Sinai University, Ismailia 41611, Egypt; mohamed.mohamed@su.edu.eg; 2Cell Biology, Histology & Genetics Division, Zoology Department, Faculty of Science, Zagazig University, Zagazig 44519, Egypt; dr.muhammadalaa@gmail.com; 3Department of Pharmacology and Toxicology, Faculty of Pharmacy, Sinai University, Ismailia 41611, Egypt; nada.elbassuony@su.edu.eg; 4Department of Microbiology and Immunology, Faculty of Pharmacy, Suez Canal University, Ismailia 41522, Egypt; ibrahimmekhimar@gmail.com; 5Department of Microbiology and Immunology, Faculty of Pharmacy, Egyptian Russian University, Cairo 11829, Egypt; daliadamsy@eru.edu.eg; 6Department of Biotechnology, Faculty of Sciences, Taif University, P.O. Box 11099, Taif 21944, Saudi Arabia; e.esmail@tu.edu.sa; 7Department of Chemistry, College of Science, Taif University, P.O. Box 11099, Taif 21944, Saudi Arabia; O.abuali@tu.edu.sa; 8Department of Medical Biochemistry, Faculty of Medicine, Zagazig University, Zagazig 44519, Egypt; nerminraafat@gmail.com; 9Department of Pathology, College of Medicine, King Khalid University, Abha 12573, Saudi Arabia; refaat_eid@yahoo.com; 10Department of Pharmaceutical Medicinal Chemistry, Faculty of Pharmacy, Horus University-Egypt, New Damietta 34518, Egypt

**Keywords:** Nipah virus, immunoinformatics, epitope mapping, multitope vaccine

## Abstract

Nipah virus is one of the most harmful emerging viruses with deadly effects on both humans and animals. Because of the severe outbreaks, in 2018, the World Health Organization focused on the urgent need for the development of effective solutions against the virus. However, up to date, there is no effective vaccine against the Nipah virus in the market. In the current study, the complete proteome of the Nipah virus (nine proteins) was analyzed for the antigenicity score and the virulence role of each protein, where we came up with fusion glycoprotein (F), glycoprotein (G), protein (V), and protein (W) as the candidates for epitope prediction. Following that, the multitope vaccine was designed based on top-ranking CTL, HTL, and BCL epitopes from the selected proteins. We used suitable linkers, adjuvant, and PADRE peptides to finalize the constructed vaccine, which was analyzed for its physicochemical features, antigenicity, toxicity, allergenicity, and solubility. The designed vaccine passed these assessments through computational analysis and, as a final step, we ran a docking analysis between the designed vaccine and TLR-^3^ and validated the docked complex through molecular dynamics simulation, which estimated a strong binding and supported the nomination of the designed vaccine as a putative solution for Nipah virus. Here, we describe the computational approach for design and analysis of this vaccine.

## 1. Introduction

Nipah virus (NiV) is a zoonotic notorious pathogen that belongs to the *Paramyxoviridae* family. The first outbreak of this virus was recorded in Malaysia about 22 years ago; following that, successive outbreaks have been reported in many countries in the south-east Asia region [1]. In a recent outbreak in 2018, the fatality rate was estimated to be 91% in a group of 23 cases in India [2]. The journey of NiV transmission to various hosts starts from the fruit bat (*Pteropus* species), which represents the viral reservoir, and when these bats drop their saliva or urine on fruits that would be consumed by humans and animals, viral infection occurs [3]. In addition to that, humans can be infected with NiV through eating NiV infected pork or even through contacting the body fluids of infected humans where man-to-man transmission occurs [4]. After the infection, cases usually suffer from mild to moderate symptoms of headache, vomiting, and fever where these symptoms may develop into severe encephalitis or acute harmful manifestations to the respiratory system, leading eventually to death [5]. Because of the high virulence, the simple way of dissemination, and the continuously elevated rates of mortality and morbidity of the viral outbreaks, NiV was categorized as a biosafety level 4 virus [6].

Exploration of the NiV genome revealed that it is composed of six essential genes, G, F, L, N, M, and P [7]. Investigation of proteins that are expressed from these genes shows that G and F proteins have a significant role in the viral entry to the infected cell, where G protein helps the virus in attachment to the host cell, then F protein guides the fusion between the viral and the host cell membranes. Translation of L protein gives rise to an RNA-dependent RNA polymerase enzyme, which is required for viral genome replication. N and M genes code for nucleoprotein and matrix protein, respectively. The distinct P gene codes for four different proteins, namely phosphoprotein, V, W, and C proteins, where the last three proteins have a vital role in fighting against the host immune response [8].

The absence of an effective drug or vaccine against NiV put this zoonotic virus on the WHO’s list for urgent need of research work to devise a solution for this emerging infectious pathogen [9]. Several approaches for designing an effective vaccine against NiV have been followed, including: vaccines based on viral vectors such as vesicular stomatitis virus [10] and rhabdovirus [11], recombinant vaccines such as recombinant measles virus vaccine, which expresses envelope glycoprotein of NiV [12], and Nipah virus-like particles composed of several NiV proteins [13].

During the last decades, there has been a great revolution in the fields of bioinformatics and structural biology with continuous updates in the computational tools for analysis of genomic data, which aided in the development of new approaches for potential vaccine design [14]. This progression led to the appearance of the immunoinformatics field, which can be defined as the interface between immunological data and computational tools that can handle these data [15]. This new approach was tried successfully against several pathogens ranging from bacteria such as *Staphylococcus aureus* [16] and *Moraxella catarrhalis* [17] to viruses such as Zika virus [18] and also fungi such as *Candida albicans* [19]. In the current study, the whole proteome of NiV was analyzed for antigenicity and virulence of each protein, where the best candidates were selected for B and T cell epitope prediction. Top ranking epitopes were assembled to construct the chimeric epitope vaccine against NiV, which was analyzed for its antigenicity and reactivity through computational tools.

## 2. Results

### 2.1. Nomination of Proteins as Vaccine Candidates

After the application of antigenicity score as the first step of NiV proteome screening, six out of nine proteins that represent the whole proteome of NiV were found to have an antigenicity score above 0.5. The functions of these six primary candidates were studied, and we selected the ones that had significant virulence roles. Consequently, we selected proteins G and F, which have roles in viral attachment to host cells, and proteins V and W that participate in combating host innate immune response. Before passing to the epitope prediction stage, we concluded that there is a high degree of similarity between the sequences of V and W proteins, therefore, we selected only protein V for epitope prediction, and one of the major criteria in the filtration of the predicted epitopes from V protein was that the epitope must show cross similarity between both V and W proteins.

### 2.2. Prediction of B Cell Epitopes

Prediction of B cell epitopes was performed through the BepiPred linear epitope prediction method (Figure 1) with a threshold value of 0.35. There were 27, 24, and 22 predicted epitopes for F, G, and V proteins, respectively. This list was downsized by selecting epitopes sized between eight to 18 peptides (Table 1 and Table 2), and, finally, we selected the top two epitopes for each protein based on the antigenicity score predicted by VaxiJen 2.0 and the conservancy analysis in the nine studied proteomes from UniProt for designing the multitope vaccine with other T cell epitopes.

### 2.3. T Cell Epitopes Prediction

Regarding MHC-I epitopes, 29,025, 32,049, and 24,165 epitopes were predicted for F, G, and V proteins, respectively, with a percentile rank from 0.01 to 100, where epitopes that had a percentile rank less than two were analyzed for the selection of best candidates, as epitopes with small percentile rank are good binders. We selected MHC-I epitopes based on their antigenicity score and the number of reactions with different alleles (Table 3 and Table 4). Moving to MHC-II epitopes, there were14,364, 15,876, and 11,934 predictions for F, G, and V proteins, respectively, and the top 10% of these predictions were analyzed according to their antigenicity score, the number of reactions with different alleles, and their ability to induce interferon-gamma (Table 5 and Table 6). Again for epitopes of V protein, we analyzed the conservancy of epitopes in W protein. As a final point before assessment of epitope candidates through docking analysis, the conservancy of the selected 18 epitopes—the total number of epitopes to be assembled into the multitope vaccine—showed 100% conservancy through multiple sequence alignment of nine targeted NiV proteomes from UniProt. Therefore, selected epitopes were predicted to possess a cross-reactivity against Nipah M and Nipah B.

### 2.4. Molecular Docking of T Cell Epitopes

Most promising MHC-I and MHC-II epitopes were analyzed for their binding affinity to a representative allele to confirm their nomination for constructing the multitope vaccine. Figure 2 shows the best docking sites of top predicted MHC-I epitopes in the receptor of HLA-A*11:01, while Figure 3 shows those of MHC-II epitopes in the receptor of HLA-DRB1*04:01. The binding energy of the docking study for suggested epitopes of each protein is shown in Table 7 where the values ranged from –7.1 to –9.1, confirming the nomination of these epitopes for constructing the multitope vaccine.

### 2.5. Multitope Vaccine Construction, Physicochemical Characteristics Assessment, and Secondary Structure Prediction

After the nomination of the best B and T cell epitopes from F, G, and V proteins, the multitope vaccine was designed based on six CTL epitopes (two from each protein), six HTL epitopes (two from each protein), and six BCL epitopes (two from each protein) linked together by GGGS, GPGPG, and KK linkers, respectively. Moreover, the beta-defensin adjuvant and the PADRE peptide sequence were also incorporated to constitute the final sequence of the designed multitope vaccine with a total length of 359 amino acids and sequenced as the following:

“EAAAKGIINTLQKYYCRVRGGRCAVLSCLPKEEQIGKCSTRGRKCCRRKKEAAAKAKFVAAWTLKAAAGGGSIVEKKRNTYGGGSTVNPSLISMGGGSEIGPKVSLIGGGSTVNPLVVNWGGGSQLDPVVTDVGGGSLSYAPEIAVGPGPGIILYVLSIASLCIGLGPGPG KYKIKSNPLTKDIVIGPGPGTLYFPAVGFLVRTEFGPGPGYQASFSWDTMIKFGDGPGPGRPGTPMPKSRGIPIKGPGPGDKLELVNDGLNIIDFKKNYNSEGIAIGKKRRVRPTSSGDKKVGQSGTCIKKLSIGSPSKIYDSKKIAVSKEDRETKKTSDDEEADQLEFKKAKFVAAWTLKAAAGGGS”.

The final predicted vaccine design was analyzed for allergenicity using the prediction approach of Blast search on allergen representative peptides (ARPs) and assessed as non-allergen. Following that, toxicity and antigenicity prediction demonstrated that the designed vaccine was non-toxic and antigenic with an antigenicity score of 0.71. Moreover, the final vaccine construction was found to be soluble upon overexpression with a SOLpro score of 0.96 (proteins with a score above 0.5 were assessed to be soluble upon overexpression). Other physicochemical characteristics of the designed vaccine, which were predicted by ProtParam online tools, are shown in Table 8. Finally, prediction of the secondary structure demonstrated that the vaccine had a 16.44% helix, a 26.46% strand, and a 57.10% coil of its secondary structure (Figure 4).

### 2.6. Tertiary Structure Prediction, Refinement, and Validation

The 3D primary structure of the predicted vaccine was modeled by 3Dpro webserver (http://scratch.proteomics.ics.uci.edu/, accessed on 20 July 2021) and validation of this structure through Ramachandran plot analysis and Z-score estimation demonstrated that 89.8%, 9.8%, and 0.4% of residues were located in favored, allowed, and outlier regions, respectively, and the estimated Z-score was –2.84. Validation of the primary structure proved the need for structure refinement, which was processed by GalaxyRefine, where the best model (Figure 5A) demonstrated an enhancement of Z-score from –2.84 to –3.17 (Figure 5B). Moreover, Ramachandran plot analysis showed an improvement for the refined structure where 93.1%, 6.6%, and 0.4% of residues were in favored, allowed, and outer regions, respectively (Figure 5C).

### 2.7. Vaccine Disulfide Engineering

After validating the refined vaccine structure, we performed disulfide engineering to increase the stability of the designed model. Regarding the current 3D structure and after analysis using the DbD2 server, 47 pairs of amino acids were found to be able to make disulfide bond, but after considering the accepted range of energy (which must be less than 2.2) and the Chi3 value (which must be between −87 and +97)), only three pairs (78ARG-84GLY), (94MET-99GLU), and (272SER-275ILE) were recommended for mutating with cysteine.

### 2.8. Molecular Docking of the Vaccine with TLR3

To validate the binding between the designed vaccine and its respective receptor through a computational approach, we ran molecular docking analysis through ClusPro 2.0 server. The server predicted 30 potential docking complexes of varying binding energies, where model number seven (Figure 6) showed the lowest binding energy value (–1263.9), which was small enough to predict a strong binding between the designed vaccine and TLR3.

### 2.9. Molecular Dynamics Simulation

We performed a molecular dynamics simulation to analyze the stability of the vaccine–receptor complex where the iMODS server was employed to run this process. The deformability of the complex relied on the individual distortion of each residue, symbolized by hinges in the chain (Figure 7B). The estimated eigenvalue of the complex was found to be 1.63 × 10^−5^ (Figure 7D). Generally, an inverse relationship was found between the eigenvalue and the variance related to each normal mode (Figure 7C). The B-factor scores of normal mode analysis in the iMODS server were equivalent to RMS (Figure 7A). The covariance matrix explained the coupling between pairs of residues where different pairs demonstrated correlated, anti-correlated, or uncorrelated motions represented by red, blue, and white colors, respectively (Figure 7E). The server formed an elastic network model (Figure 7F), as it showed the pairs of atoms linked through springs according to the degree of stiffness between them, and that was represented by color where it moved to darker gray with stiffer strings.

### 2.10. Vaccine Reverse Translation and Codon Optimization

The last stage of the current computational analysis was reverse translation and codon optimization on the amino acid sequence of the multitope vaccine where the JCat server was employed for this process. Regarding the submitted amino acid sequence of the multitope vaccine, the measured GC content was 50.23%, which was accepted as it was located within the accepted range (30%–70%). Furthermore, the Codon Adaptation Index (CAI) was calculated as 0.99, demonstrating a high probability of protein expression when we transferred our experiments to the wet lab; the value of CAI ranged 0–1, and the accepted range is between 0.8 and 1.

## 3. Discussion

Nipah virus is an emerging zoonotic pathogen with a wide range of pathogenesis and a high percentage of fatality [20]. The antiviral drug, ribavirin, was used with the early diagnosed cases with NiV, where it showed some improvement with these cases, but the fatality rate was still high [21]. Moreover, to assess the activity of ribavirin against NiV, it was administered along with chloroquine to Nipah infected hamster models, where they failed to prevent animal death [22]. With successive outbreaks of NiV and the absence of an effective drug that can cure patients and reduce the high fatality rate of NiV, the development of an effective vaccine became a significant health priority. During the last decade, several trials that adopted different approaches were performed to provide a putative solution for this deadly virus. One of the early trials was performed in 2012 based on Hendra virus G glycoprotein to design a subunit vaccine against NiV [23]. Unfortunately, a detailed analysis of that promising vaccine showed 100% relapse of encephalitis on tested African green monkeys [24]. Other vector and virus-like particle vaccines were designed, but, until now, none of these vaccines have been approved for human usage.

Recently, there has been a revolution in the field of vaccine development as a result of the great progression in bioinformatics, structural biology, and computational tools that aided largely in the process of handling and analyzing genomic data of several microorganisms [25]. The approach of predicting and designing vaccines through in silico studies has improved massively in the last few years, where its applications extended to involve bacteria, viruses, fungi, and even cancer [26]. Multitope vaccine has been predicted through computational approaches against several microorganisms such as Mayaro virus [27], Lassa virus [28], COVID-19 [29], and *E. coli* [30], where the predicted vaccine of the last study was expressed and analyzed through wet lab experimental validation and showed protection against urinary tract infection caused by uropathogenic *E. coli* in animal models. The application of immunoinformatics for designing NiV vaccine through epitope prediction was shown in many studies. In 2014, a study predicted B and T cell epitopes of G and F proteins and analyzed their probability to act as vaccine candidates [31]. With a similar approach, only T cell epitopes of all NiV proteins were predicted for designing peptide-based vaccines [32]. Moreover, B and T cell epitopes of membrane proteins and RNA-dependent RNA polymerase (RdRp) proteins were also analyzed in the studies [33] and [34], respectively. The commonality between these studies is that proteins under investigation were selected without a clear rationale, and the candidates were single epitopes. In addition to that, some of these studies reported vaccines based on a single viral protein, and RNA viruses have a high mutation rate, as continuous mutations contribute largely to viral pathogenicity and early ineffectiveness of vaccine. This is a major disadvantage of a single viral protein vaccine [35].

On the other hand, in the current study, we selected our protein candidates after the analysis of the whole proteome for the antigenicity score and the virulence role of every single protein. The primary list was composed of four proteins, G, F, V, and W, which had a high antigenicity score and possessed a vital role in NiV virulence. However, after careful analysis of the amino acid sequence of proteins V and W, we found that there was a high similarity in their sequences; therefore, we excluded W protein and used the factor of epitope cross-match between V and W proteins as a major one for selecting the most prominent epitopes from V protein. G protein has a vital role in the attachment of NiV to two cellular receptors, ephrin-B2 and ephrin-B3, and this subsequently triggers F-mediated membrane fusion between NiV and the infected cell [36]. Viral surface proteins are potential targets for vaccine design [37]. Moving to the contribution of V and W proteins in NiV virulence, it is known that signal transducers and activators of transcription (STAT) proteins are transcription factors that play key roles in interferon (IFN) and cytokine signaling. V and W proteins have a common N-terminal sequence that binds to STAT1 and STAT2 and blocks IFN-induced signal transduction, allowing the virus to evade the human immune response [38].

Another major advantage of the current study is that the vaccine was constructed based on multitopes, which have superior efficacy and protection against infectious agents [39] over a single epitope-based vaccine. Therefore, the current study did not end solely with prediction of B and T cell epitopes as did the above-mentioned studies; instead, top-ranked epitopes were selected to construct the multitope vaccine. Ranking criteria relied on percentile rank, antigenicity, ability to induce interferon-gamma, binding affinity to a representative allele, the conservancy of these epitopes (because, as we mentioned, NiV has a high rate of mutation), and the number of reactive alleles with the selected epitope to cover a high percentage in terms of population coverage. As mentioned, top-ranked epitopes were arranged together using appropriate linkers. Moreover, the beta-defensin adjuvant and the PADRE peptide were also incorporated into the final construct of the multitope vaccine to strengthen the stimulated immune response and reduce the HLA polymorphism in the population [40].

Before predicting the three-dimensional structure of the designed multitope vaccine, the final vaccine amino acid sequence was analyzed regarding its physicochemical properties, antigenicity, allergenicity, solubility upon over-expression, and toxicity using computational tools. The analysis demonstrated that the proposed sequence of the vaccine is stable, hydrophilic, soluble upon over-expression, antigenic, non-allergen, and non-toxic. The next step was predicting the 3D structure, where structure validation using Ramachandran plot and ProSA demonstrated that a good-quality 3D structure was modeled and, for a better design, protein refinement was performed. The final refined structure was assessed, which proved generation of a high-quality 3D model. To provide more stability to the final designed model, we performed disulfide engineering, as disulfide bonds were proved to reduce the conformational entropy and increase the free energy of the denatured state of the designed protein, leading to an elevation in the stability of protein conformation [41]. Finally, molecular docking between the designed vaccine and TLR-3 was performed to analyze the binding affinity between the complex components, where the docking score of the generated complex revealed that there was a good affinity between the vaccine and its receptor. To obtain a closer view of the docked complex, we employed normal mode analysis that was integrated into iMODS server and the output data, which described the collective functional motions of the complex and predicted that the designed vaccine can stimulate a specific immune response against NiV.

## 4. Materials and Methods

An overview of the flow of work is shown in Figure 8.

### 4.1. Selection of Vaccine Candidates

One of the standard proteomes of the Nipah virus was retrieved from UniProt with proteome Id (UP000002330). The complete proteome was analyzed to detect proteins with antigenicity scores of more than 0.5 that have a role in the virus pathogenicity to be nominated as the vaccine candidates of the current study. The antigenicity score was calculated via Vaxigen v2.0 [42]. Investigation of Nipah virus proteomes on UniProt showed that a total of 15 proteomes were uploaded and, out of them, three proteomes were standard and six proteomes were close to standard. These nine proteomes cover the two major subtypes of the Nipah virus (Nipah M from Malaysia and Nipah B from Bangladesh). The protein sequence of the vaccine candidates was obtained from these nine proteomes to run multiple sequence alignment to confirm the conservation of the selected epitopes in stage two of the current study.

### 4.2. Prediction of B and T Cell Epitopes

Filtered proteins were submitted to the Immune Epitope Database (IEBD) [43]. MHC-I binding predictions were performed through NetMHCpan EL 4.0 prediction method, the default prediction method recommended by the server. The reference set of HLA alleles was used with this prediction method, as it acts as a representative for commonly shared binding specificities and gives more than 97% in terms of population coverage [44]. Moving to the prediction of MHC-II binding, the IEBD-recommended 2.22 prediction method along with the full HLA reference set, which covers more than 99% in terms of population coverage [45], were applied, and for those epitopes that had the best scores, the ability for inducing interferon-gamma was predicted through INF prediction server [46]. Finally, prediction of B cell epitopes was performed through the BepiPred linear epitope prediction method [47], which runs based on a combination approach of a hidden Markov model and a propensity scale method. Before moving to the next stage of constructing the multitope vaccine, top-ranked epitopes were analyzed for their conservancy through multiple sequence alignment and for their reactivity through molecular docking, where the 3D structure for each epitope was predicted through PEP FOLD 3 webserver [48]. The crystal structures of HLA-A*11:01 (PDB ID 6JP3) and HLA-DRB1*04:01 (PDB ID 5JLZ) were chosen for MHC-I and MHC-II epitopes, respectively, as a receptor for molecular docking. Docking analysis was performed through AutoDock Vina [49].

### 4.3. Multitope Vaccine Construction

There were four components merged for the construction of the multitope vaccine. Firstly, we introduced the β-defensin adjuvant. Following that, selected epitopes from the previous steps were included and linked together with GGGS, GPGPG, and KK linkers for CTL, HTL, and BCL epitopes, respectively. Finally, the PADRE sequence, which was found to improve the immune response for the designed vaccine [50], was incorporated. After completion of the vaccine construction, it was assessed for its antigenicity through VaxiJen v 2.0 [42] and for its allergenicity through AlgPred [51]. Finally, the toxicity of the designed vaccine was predicted through the ToxinPred web server [52].

### 4.4. Prediction of Protein Solubility, Physicochemical Characters, and Secondary Structure Assessment

Protein solubility upon overexpression in *Escherichia coli* was predicted through the SOLpro server through a two-stage SVM architecture based on multiple representations of the primary sequence [53]. Vaccine physicochemical characters such as molecular weight, atomic composition, instability index, in addition to many other characteristics were predicted through the ProtParam tool [54]. Finally, the secondary structure prediction of the constructed vaccine was performed through the PSIPRED server, which incorporates two feed-forward neural networks performing an analysis on output obtained from PSI-BLAST [55].

### 4.5. Vaccine 3D Structure Prediction and Validation

Vaccine tertiary structure was predicted through a 3Dpro web server [56]. This server constructs multiple models using random seeds and gives an outcome of the best-predicted model based on energy. Therefore, the predicted vaccine structure would have the lowest possible energy and the highest stability. Before validating the generated 3D model, protein refinement was performed through the GalaxyRefine server [57]. The refinement process occurs through rebuilding and repacking of the side chains followed by structure relaxation through molecular dynamics simulation. Validation of the 3D structure before and after refinement gives an overview of structure improvement and its current quality. Validation of the vaccine 3D structure was performed through Ramachandran plot analysis [58] and ProSA [59].

### 4.6. Vaccine Disulfide Engineering

Disulfide bonds have a significant role in enhancing protein stability and improving its geometric conformation. Therefore, before moving to the docking analysis, Disulfide by Design 2.0 web server [60] was employed to sign disulfide bonds to the predicted 3D structure of the vaccine.

### 4.7. Docking Analysis between Predicted Vaccine 3D Structure and TLR-3

A docking study was performed to estimate potential binding orientations between the ligand and its prospective receptor, where the generated binding energy could give an overview of the binding affinity between the components of that complex [61]. NiV infection was found to induce TLR-3 expression, as it specifically identified the dsRNA of the virus [62]. Thus, TLR-3 (PDB id: 1ZIW), which acted as a receptor, was obtained from the protein data bank and directly submitted to the ClusPro 2.0 server [63] along with the refined 3D structure of the designed vaccine, which acted as a ligand. The server performed the docking analysis by making a large number of possible conformations, then clustering 1000 structures with the lowest energy and finally removing steric clashes.

### 4.8. Molecular Dynamics Simulation

To study molecular behavior and assess the stability of the protein–ligand complex, molecular dynamics simulation was applied, as it gives an overview of the physical basis of the analyzed complex [64]. In the current study, we used the iMODS server [65] for this kind of analysis of the complex between the designed vaccine 3D structure and TLR-3. This server has the advantage of being fast with accurate estimations; it analyzed the collective motions of the vaccine–receptor complex using normal mode analysis in internal coordinates [66].

### 4.9. Reverse Translation and Codon Adaptation

As a final point of our computational study, we performed codon adaptation for the designed vaccine in a proposed expression host to prepare for wet lab production of the designed vaccine as a prospective stage. We selected *E. coli* k-12 as an expression host. It is known that there are no similarities between the codon usage of humans and *E. coli*, hence, this step is a major one to achieve a high rate of wet lab protein expression in the detected host. We selected the JCAT server [67] for this step. It calculates the Codon Adaptation Index (CAI) based on an algorithm, where each codon is given weight to the subset of highly expressed genes defined for the considered organism, and if there is no adaptation required, the CAI value is 1.0.

## 5. Conclusions

The application of computational approaches for predicting and validating designed vaccines against various pathogens is a promising technique with a great economical value, as it shortens many experimental steps when we move to wet lab validation. In the current study, three antigenic virulent proteins were selected to act as candidates for designing a subunit vaccine against NiV where a novel multitope vaccine was constructed using top-ranked B and T cell epitopes from nominated proteins along with suitable linkers, adjuvant, and PADRE sequence. The designed vaccine was analyzed, and the analysis output predicted that the designed vaccine had optimum physicochemical, structural, and immunological characteristics, which enabled it to stimulate an appropriate immune response against NiV. Finally, experimental trials are required to prove the practical efficacy of this potential vaccine construct.

## Figures and Tables

**Figure 1 ijms-22-09330-f001:**
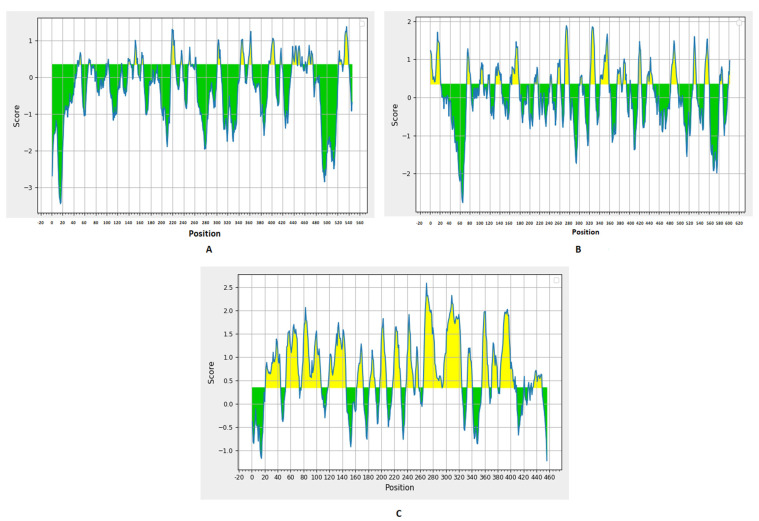
BepiPred linear epitope prediction for F (**A**), G (**B**), and V (**C**) proteins; the yellow section represents the B epitope part of each protein.

**Figure 2 ijms-22-09330-f002:**
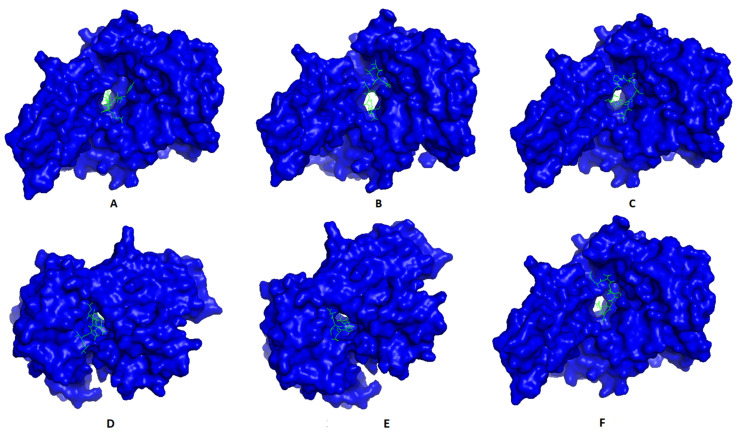
Structural positions of MHC-I epitopes (green color) in three-dimensional structure of HLA-A*11:01 (blue color); structures **A, B, C, D, E**, and **F** are for epitopes numbers 1, 2, 3, 4, 5, and 6, respectively, from Table 7.

**Figure 3 ijms-22-09330-f003:**
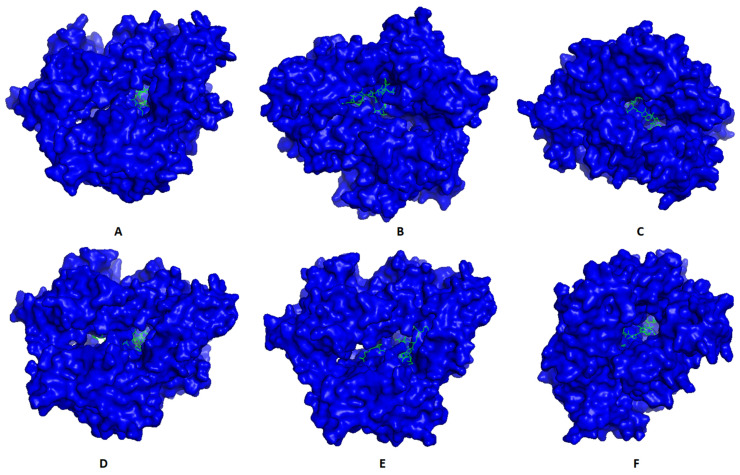
Structural positions of MHC-II epitopes (green color) in three-dimensional structure of HLA-DRB1*04:01 (blue color); structures **A, B, C, D, E**, and **F** are for epitopes numbers 1, 2, 3, 4, 5, and 6, respectively, from Table 7.

**Figure 4 ijms-22-09330-f004:**
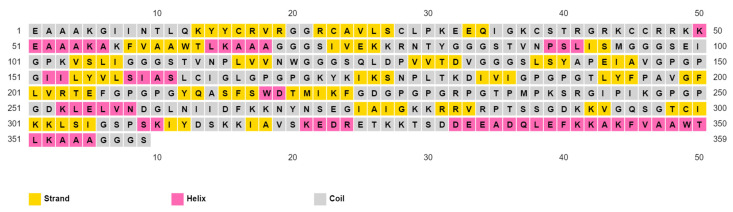
Secondary structure prediction of designed multitope vaccine using PESIPRED server.

**Figure 5 ijms-22-09330-f005:**
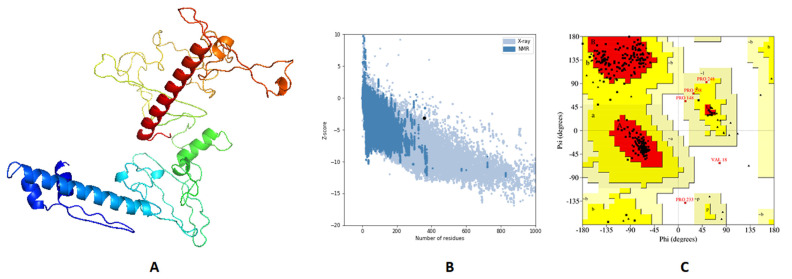
Structural analysis of the predicted vaccine 3D structure. (**A**) The 3D structure of designed vaccine after refinement; (**B**) ProSA-web evaluation of the refined vaccine where the black dot shows the exact Z-score; (**C**) Ramachandran plot analysis of the refined vaccine.

**Figure 6 ijms-22-09330-f006:**
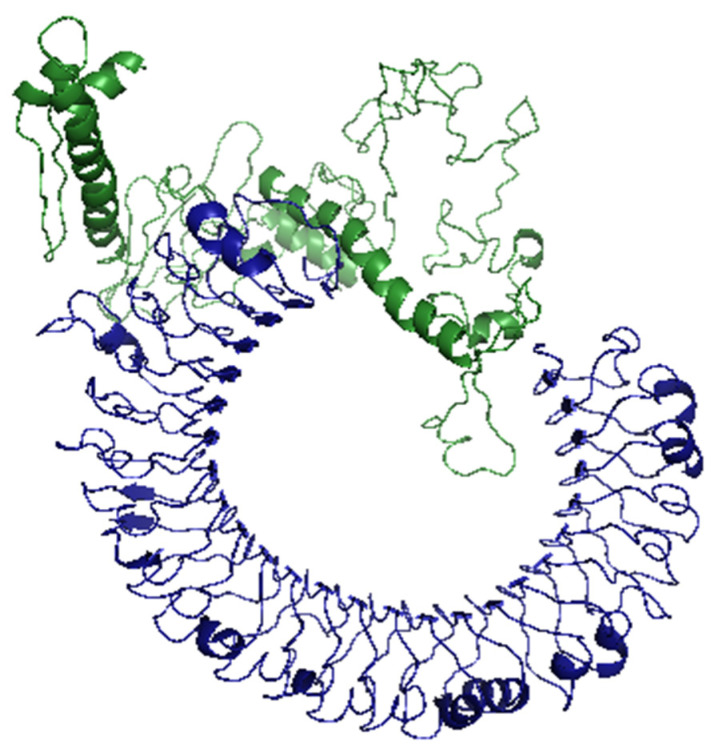
Docked complex of the designed vaccine (green color) and TLR3 (blue color).

**Figure 7 ijms-22-09330-f007:**
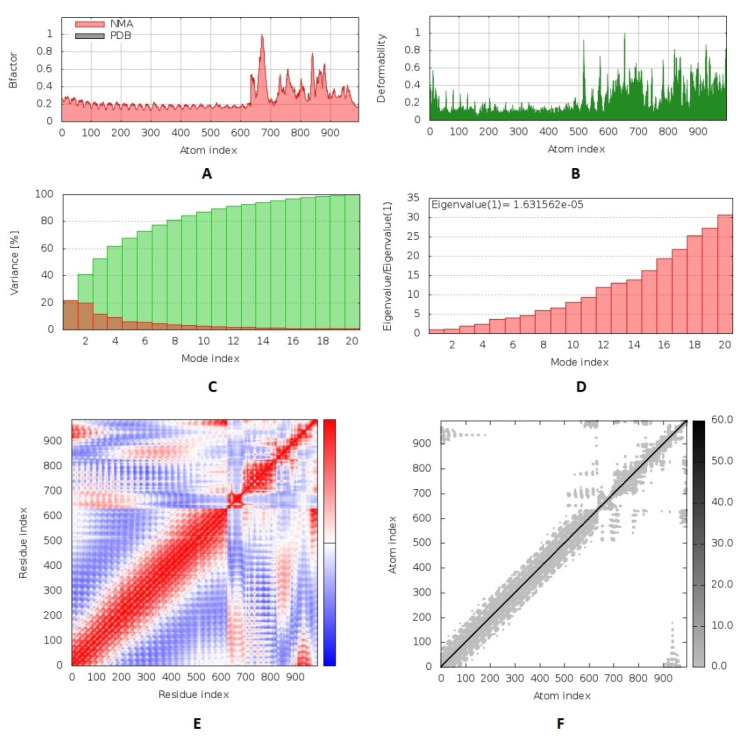
Molecular dynamics simulation of the predicted vaccine–TLR3 complex; complex stability was assessed through B-factor values (**A**), deformability (**B**), variance (**C**), eigenvalue (**D**), the covariance of residue index (**E**), and elastic network analysis (**F**).

**Figure 8 ijms-22-09330-f008:**
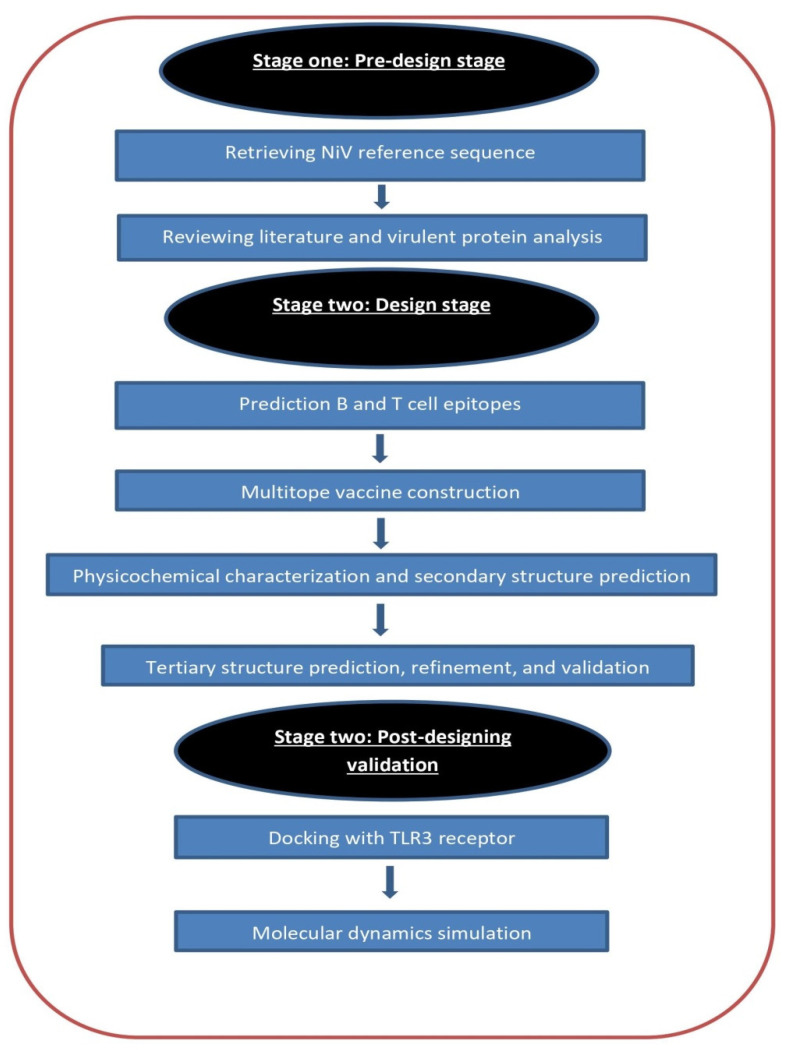
Graphical representation of the applied strategy for designing a chimeric epitope vaccine against Nipah virus.

**Table 1 ijms-22-09330-t001:** Predicted B cell epitopes from F and G proteins.

F			G		
Epitope	Start-End	Antigenicity Score	Epitope	Start-End	Antigenicity Score
NLQDPVSNS	217–225	0.01	TRSTDNQAV	74–82	0.45
DYATPMTN	243–250	0.18	ISQSTASINENVN	131–143	0.23
QTTGRAISQSGE	395–406	0.51	PLPFREYRPQTEGVS	165–179	0.94
NYNSEGIAIG	437–446	1.24	VGQSGTCI	210–217	0.62
QSLQQSKDYIK	465–475	0.18	WTPPNPNTV	271–279	0.67
RRVRPTSSGD	531–540	1.28	KPKSNGGGYNQ	322–332	0.49
			KGRYDKVMPYGPSGIKQG	342–359	0.7
			KYNDSNCPI	376–384	0.81
			CQYSKPEN	387–394	–0.13
			LSIGSPSKIYDS	436–447	0.4
			VISRPGQSQCPRFN	484–497	–0.18
			ASEDTNAQK	552–560	0.23

**Table 2 ijms-22-09330-t002:** Predicted B cell epitopes from V protein.

Epitope	Start-End	Antigenicity Score	Crossmatch with Protein W
IAVSKEDRET	163–172	1.11	Yes
STTGLNPTA	182–190	1.33	Yes
LSDPAKDSPVI	198–208	0.12	Yes
KEQNVGPQTSRNV	218–230	1.32	Yes
TSDDEEADQLEF	239–250	0.61	Yes
ECSGSEDP	333–340	–0.22	Yes
QGKDAQPPYHW	356–366	1.21	Yes
SISPDKTEIV	371–380	0.96	Yes
TPLPRRQECQCGEC	435–448	0.46	No

**Table 3 ijms-22-09330-t003:** Top-ranked T cell epitopes (MHC-I peptides) of F and G proteins.

F			G		
Epitope	Antigenicity	Reacting Alleles Number	Epitope	Antigenicity	Reacting Alleles Number
IVEKKRNTY	1.54	8	VGFLVRTEF	1.83	9
TVNPSLISM	1.24	19	FLVRTEFKY	1.48	8
EWISIVPNF	1.06	7	EIGPKVSLI	1.37	9
IQELLPVSF	0.97	11	ISCPNPLPF	0.97	10
AQITAGVAL	0.79	9	TVNPLVVNW	0.92	14
RPTSSGDLYY	0.78	11	ILKPKLISY	0.83	19
LSLDLALSKY	0.77	7	RGDEVPSLF	0.8	9
SECSVGILHY	0.69	6	AMDEGYFAY	0.72	14
YLSDLLFVF	0.68	15	LAMDEGYFAY	0.65	8
NVIISLGKY	0.66	6	SFSWDTMIKF	0.58	7

**Table 4 ijms-22-09330-t004:** Top-ranked T cell epitopes (MHC-II peptides) of F and G proteins.

F				G			
Epitope	Antigenicity	IFN Epitope	Reacting Alleles	Epitope	Antigenicity	IFN Epitope	Reacting Alleles
LITFISFIIVEKKRN	1.52	No	11	PAVGFLVRTEFKYND	1.49	Yes	6
IGLITFISFIIVEKK	1.44	No	10	AVGFLVRTEFKYNDS	1.41	Yes	5
GLITFISFIIVEKKR	1.42	No	11	FPAVGFLVRTEFKYN	1.38	Yes	7
CIGLITFISFIIVEK	1.15	No	11	YFPAVGFLVRTEFKY	1.38	Yes	7
IILYVLSIASLCIGL	1.02	No	10	NPKVVFIEISDQRLS	1.26	Yes	6
MAGVAIGIATAAQIT	0.99	No	10	LYFPAVGFLVRTEFK	1.25	Yes	7
ISLGKYLGSVNYNSE	0.66	Yes	3	NPLVVNWRNNTVISR	1.04	Yes	6
KYKIKSNPLTKDIVI	0.62	Yes	3	TLYFPAVGFLVRTEF	1.00	Yes	8
AQRLLDTVNPSLISM	0.6	Yes	3	YQASFSWDTMIKFGD	0.99	Yes	8
GKYLGSVNYNSEGIA	0.57	Yes	3	ENPKVVFIEISDQRL	0.91	Yes	7

**Table 5 ijms-22-09330-t005:** Top-ranked T cell epitopes (MHC-I peptides) of V protein.

Epitope	Antigenicity	Reacting Alleles Number	Crossmatch with Protein W
GLNPTAVPF	1.71	8	Yes
ICWDGKRAW	1.6	6	No
NPTAVPFTL	1.54	6	Yes
KSRGIPIKK	1.39	6	Yes
KDAQPPYHW	1.23	8	Yes
DQLEFEDEF	1.03	7	Yes
AQPPYHWSI	0.91	13	Yes
SIKDQTKAW	0.9	10	Yes
QLDPVVTDV	1.24	5	Yes
LSYAPEIAV	0.86	5	Yes

**Table 6 ijms-22-09330-t006:** Top-ranked T cell epitopes (MHC-II peptides) of V protein.

Epitope	Antigenicity	IFN Epitope	Reacting Alleles Number	Crossmatch with Protein W
RETDLVHLENKLSTT	1.4	Yes	8	Yes
VIAEHYYGLGVKEQN	1.37	Yes	6	Yes
LVHLENKLSTTGLNP	1.36	Yes	5	Yes
DLVHLENKLSTTGLN	1.32	Yes	8	Yes
TDLVHLENKLSTTGL	1.19	Yes	8	Yes
ETDLVHLENKLSTTG	1.16	Yes	8	Yes
GNVCLVSDAKMLSYA	0.77	Yes	16	Yes
PYHWSIERSISPDKT	0.65	Yes	4	Yes
RPGTPMPKSRGIPIK	1.08	Yes	1	Yes
DKLELVNDGLNIIDF	0.81	No	11	Yes

**Table 7 ijms-22-09330-t007:** The binding energy of T cell epitopes with their respective allele.

No.	Epitope	MHC-I Allele	Binding Energy (kcal/mol)	Epitope	MHC-II Allele	Binding Energy (kcal/mol)
1	IVEKKRNTY		–8.0	IILYVLSIASLCIGL		–7.8
2	TVNPSLISM		–7.1	KYKIKSNPLTKDIVI		–7.5
3	EIGPKVSLI	HLA-A*11:01	–7.2	TLYFPAVGFLVRTEF	HLA-DRB1*04:01	–9.0
4	TVNPLVVNW		–9.1	YQASFSWDTMIKFGD		–8.2
5	QLDPVVTDV		–8.3	RPGTPMPKSRGIPIK		–8.1
6	LSYAPEIAV		–8.4	DKLELVNDGLNIIDF		–7.4

**Table 8 ijms-22-09330-t008:** Physicochemical characteristics of the designed multitope vaccine.

Physicochemical Characteristic	Molecular Weight	Theoretical pI	Extinction Coefficient	GRAVY	Instability Index	Aliphatic Index
Score	37.29 kDa	9.73	37,400 M^−1^ cm^−1^	–0.281	29.40	77.14

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
