# Peer review of "Proteome Based Approach Defines Candidates for Designing a Multitope Vaccine against the Nipah Virus"

_ijms, 2021, doi:10.3390/ijms22179330_

Round 1
Reviewer 1 Report
The authors of this paper have used in silico methods to design a novel multivalent vaccine against Nipah virus, which currently lacks any clinical approved treatment or vaccine. The study design is very rationale and the results are well organized and communicated. It will be interesting to see how well in silico predictions work in animal models.
As a virologist, though, I have a few comments and questions ...
- why does the vaccine dock with TLR3? vaccines don't normally have receptors.
- could you please explain why the virulence of the viral protein plays a role in the selection for vaccine epitopes? while glycoproteins make good candidates for b-cell epitopes, t-cell epitopes can be in any protein as long as they are displayed by MHC molecules and have high antigenicity. in addition, some "non-virulent" viral proteins like nucleoproteins and polymerases are highly conserved and make good targets for epitopes that provide protection against multiple strains of the same virus. i would advise including these in your current and future vaccines if they have good scores.
- most importantly, there are two different subtypes of nipah - nipah m (from malaysia) and nipah b (from bangladesh). at the very least, you need to indicate which one you are designing your vaccine against. and since you are already doing comprehensive computer predictions, why not show whether a vaccine can be created that would be protective against both? i.e. find areas of conservation, or include epitopes specific against both nipah-m and nipah-b.
- is anything known about whether the order of epitopes affects efficacy? does switching the order significantly change the structure? this could be addressed in the discussion if there is any literature available.
- in your discussion please make reference to other studies that have designed vaccines that are epitopes with linkers. have any been tested in animal models and shown efficacy?
Author Response
Comments and Suggestions for Authors
The authors of this paper have used in silico methods to design a novel multivalent vaccine against Nipah virus, which currently lacks any clinical approved treatment or vaccine. The study design is very rationale and the results are well organized and communicated. It will be interesting to see how well in silico predictions work in animal models.
Answer: The authors thank the reviewer for his positive impact on our study.
As a virologist, though, I have a few comments and questions ...
- why does the vaccine dock with TLR3? vaccines don't normally have receptors.
Answer: We considered the docking of the mutitope vaccine with TLR3, which showed overexpression with Nipah virus-infected cases, to show that the designed vaccine would trigger a specific response against Nipah virus. Kindly refer to studies 27, 28, 29, 30 in the reference list which applied a similar approach against other pathogens with different TLR specific activation.
could you please explain why the virulence of the viral protein plays a role in the selection for vaccine epitopes? while glycoproteins make good candidates for b-cell epitopes, t-cell epitopes can be in any protein as long as they are displayed by MHC molecules and have high antigenicity. in addition, some "non-virulent" viral proteins like nucleoproteins and polymerases are highly conserved and make good targets for epitopes that provide protection against multiple strains of the same virus. i would advise including these in your current and future vaccines if they have good scores.
Answer: The authors thank the reviewer for his advice and we will consider that in our future work. Just to clarify, we added the protein virulence filtration step to shorten the list of protein candidates with high antigenicity and consequently design a multitope vaccine with an appropriate size. Also, because virulent proteins are important targets for recombinant vaccine design and have been selected in many previous trials as vaccine candidates we put this filtration step.
- most importantly, there are two different subtypes of nipah - nipah m (from malaysia) and nipah b (from bangladesh). at the very least, you need to indicate which one you are designing your vaccine against. and since you are already doing comprehensive computer predictions, why not show whether a vaccine can be created that would be protective against both? i.e. find areas of conservation, or include epitopes specific against both nipah-m and nipah-b.
Answer: Thank you for your deep observation. In our study, the nine selected proteomes cover both types of Nipah virus and the assembled epitopes in the multitope vaccine were 100% conserved through the multiple sequence alignment of these nine proteomes therefore the vaccine candidate of the current study would cover both types of Nipah virus. Kindly refer to updated lines (134-135) and (355-357).
- is anything known about whether the order of epitopes affects efficacy? does switching the order significantly change the structure? this could be addressed in the discussion if there is any literature available.
Answer: After investigation, we did not find any study that analyzed the point of switching the order of peptides. Studies only analyzed the binding affinity of the single epitopes and/or the multitope vaccine and we discussed that in the discussion section.
- in your discussion please make reference to other studies that have designed vaccines that are epitopes with linkers. have any been tested in animal models and shown efficacy?
Answer: Ok, references have been added as requested. Yes, multitope vaccine against E. coli was expressed and showed protection in animal models. Kindly refer to lines (275-280)
Reviewer 2 Report
The manuscript entitled "Proteome based approach defines candidates for designing a multitope vaccine against Nipah virus" by Soltan and co-authors Soltan describe a computational approach to design a protective vaccine against the Nipah virus. The authors have analysed the Nipah virus proteome searching for antigenicity and virulence role of each protein, concluding that proteins F, G, V and W are the candidates for epitope prediction. Several cmputational tools were used to design a recombinant protein that was inspected by bioinformatics tools for physicochemical features, antigenicity, toxicity, allergenicity, and solubility. Docking studies were also conducted through molecular dynamics simulation which estimated a strong binding of the protein to selected virus receptors. The work is well organized and described, with a few minor English corrections required. However, a major issue is that no experimental work was performed to corroborate any of the bioinformatics findings. For instance, none of the predicted immunogenic epitopes was demonstrated to be immunogenic. The construction of a recombinant plasmid harboring the engineered protein and the demonstration that the protein is expressed would give further strength to the work.
Othe rminor issues:
line 85: substitute " is planned to be" by "was".
line 87: substitute "will be" by "was".
line 88: substitute "will be" by "was".
line 98: substitute "investigated" by "conclude".
Author Response
The manuscript entitled "Proteome based approach defines candidates for designing a multitope vaccine against Nipah virus" by Soltan and co-authors Soltan describe a computational approach to design a protective vaccine against the Nipah virus. The authors have analysed the Nipah virus proteome searching for antigenicity and virulence role of each protein, concluding that proteins F, G, V and W are the candidates for epitope prediction. Several cmputational tools were used to design a recombinant protein that was inspected by bioinformatics tools for physicochemical features, antigenicity, toxicity, allergenicity, and solubility. Docking studies were also conducted through molecular dynamics simulation which estimated a strong binding of the protein to selected virus receptors. The work is well organized and described, with a few minor English corrections required.
Answer: The authors thank the reviewer for his positive impact on our study.
However, a major issue is that no experimental work was performed to corroborate any of the bioinformatics findings. For instance, none of the predicted immunogenic epitopes was demonstrated to be immunogenic. The construction of a recombinant plasmid harboring the engineered protein and the demonstration that the protein is expressed would give further strength to the work.
Answer: The authors agree with the reviewer at this point as the work is based on a computational approach and we want to clarify some points:
- The authors already mentioned in the abstract and the conclusion sections that it is a computational approach and experimental trials are required to prove the practical efficacy of this potential vaccine construct.
- There were many findings in the study and it would be too long to add more data about protein expression and testing.
- One of the main aims of the current study is to demonstrate the value of bioinformatics tools in saving both cost and time therefore we applied the computational approach to confirm that bioinformatics tools can generate valuable results in the field of vaccine development.
- There are several published articles in highly reputed journals, including IJMS, based only on the computational approach. For examples kindly refer to (https://doi.org/10.3390/ijms18020371), (https://doi.org/10.1038/s41598-019-49354-z) and (https://doi.org/10.1080/07391102.2020.1780944).
- The most important point is that authors have a plan to transfer the current study results to wet-lab experiments to validate these results and we are currently working on that.
Other minor issues:
line 85: substitute " is planned to be" by "was".
line 87: substitute "will be" by "was".
line 88: substitute "will be" by "was".
line 98: substitute "investigated" by "conclude".
Answer: Ok, these modifications were done inside the manuscript as requested.
Round 2
Reviewer 1 Report
Thank you for your replies and best of luck with your vaccine in future studies.
Reviewer 2 Report
The manuscript was revised taking into consideration the minor changes suggested. I still miss the experimental demonstration of some findings.